# Cancers: What Are the Costs in Relation to Disability-Adjusted Life Years? A Systematic Review and Meta-Analysis

**DOI:** 10.3390/ijerph19084862

**Published:** 2022-04-16

**Authors:** Jacopo Garlasco, Mario Cesare Nurchis, Valerio Bordino, Martina Sapienza, Gerardo Altamura, Gianfranco Damiani, Maria Michela Gianino

**Affiliations:** 1Department of Public Health Sciences and Paediatrics, University of Turin, 10126 Turin, Italy; jacopo.garlasco@unito.it (J.G.); valerio.bordino@unito.it (V.B.); mariola.gianino@unito.it (M.M.G.); 2Department of Woman and Child Health and Public Health, Fondazione Policlinico Universitario A, Gemelli IRCCS, 00168 Rome, Italy; gianfranco.damiani@unicatt.it; 3Department of Health Sciences and Public Health, Università Cattolica del Sacro Cuore, 00168 Rome, Italy; martina.sapienza93@gmail.com (M.S.); gerardoaltamura@outlook.it (G.A.)

**Keywords:** cost-per-DALY ratio, health policy, cancer, chronic diseases, systematic review

## Abstract

Cancers currently represent a leading cause of morbidity and mortality, and precisely estimating their burden is crucial for evidence-based decision-making. This study aimed at understanding the average costs of cancer-related disability-adjusted life years (DALYs) and highlighting possible differences in economic estimates obtained with diverse approaches. We searched four scientific databases to identify all the primary literature simultaneously investigating cancer-related costs and DALYs. In view of the different methodologies, studies were divided into two groups: those estimating costs starting from DALYs, and those independently performing cost and DALY analyses. The latter were pooled to compute costs per disease-related DALY: meta-analytic syntheses were performed for total costs and indirect costs, and in relation to the corresponding gross domestic product (GDP) per capita. The quality of included studies was assessed through the Quality of Health Economic Studies instrument. Seven studies were selected. Total and indirect pooled costs per DALY were, respectively, USD 9150 (95% CI: 5560–15,050) and USD 3890 (95% CI: 2570–5880). Moreover, the cost per cancer-related DALY has been found to be, on average, 32% (95% CI: 24–42%) of the corresponding countries’ GDP per capita. Costs calculated a priori from DALYs may lead to results widely different from those obtained after data retrieval and model building. Further research is needed to better estimate the economic burden of cancer in terms of costs and DALYs.

## 1. Introduction

Cancer is widely recognized as a global problem for incidence, mortality, years of life lost and years lived with disability. Worldwide, an estimated 19.3 million new cancer cases (18.1 million excluding non-melanoma skin cancer) and almost 10.0 million cancer deaths (9.9 million excluding non-melanoma skin cancer) occurred in 2020 [1]. Furthermore, in 2017 cancer caused a global burden of 226.5 million years of life lost and 7 million years lived with disability [2]. Cancer causes health, financial and socioeconomic problems (including expenditure for medical and non-medical services, loss of productivity, and disability), thereby having an impact at an individual, family and social level [3].

Measuring the burden of disease related to cancer is a crucial topic for a public health system; as such, an evaluation is fundamental to assess the health status of a country and to make comparisons between countries possible [4]. The importance of establishing the disease burden related to cancer as precisely as possible is also dictated by the need to properly allocate the resources available to the national health system for the treatment and prevention of this diverse group of diseases [5].

Previous studies have used disability-adjusted life years as a measure of the epidemiologic burden of different cancers [6]. The disability-adjusted life-year (DALY) is a health metric combining the morbidity and mortality of a disease; it represents the sum of years lived with disability (YLD) and years of life lost (YLL) [7]. Expressing disease burdens in DALYs allows for the comparison of different diseases and therefore represents a useful tool for evidence-based healthcare policy-making [8].

The most recent estimate provided by the World Health Organization stated that the total annual economic impact of cancer worldwide is USD 1.16 trillion: this figure represents around 1.5% of the world’s gross domestic product (GDP), amounting to approximately USD 85 trillion [9]. National costs for cancer care in the United States (U.S.) were estimated to be USD 190.2 billion in 2015 and USD 208.9 billion in 2020, with a 10% increase that was found to be primarily justified by the aging and growth of the U.S. population [10]. Other analyses estimated a direct health expenditure of around EUR 58 billion in five European countries only (France, Germany, Italy, Spain and United Kingdom) [11].

Accurately estimating the economic value of cancer-related DALYs should be an important goal for healthcare systems, especially considering that currently 30–50% of cancers can be prevented by avoiding risk factors and by implementing the existing evidence-based prevention strategies [12]. The cancer burden can also be reduced through early cancer detection, and appropriate treatment and care of patients who develop cancer [12]. Evaluating the economic burden of cancers would thus lead to better understand the potential economic benefits of such prevention or therapy strategies.

However, the exact economic impact of cancer is not completely understood because multiple methodologies have been adopted to estimate the value of DALYs ascribable to cancer, sometimes relying on the assumption that each DALY could be a priori valued at one year’s GDP per capita of the nation under investigation [13].

With that in mind, this study aimed at assessing the average cost per DALY of cancer by reviewing the currently available literature. As different methodologies have been used over time to evaluate the health and economic burden attributable to specific types of cancer worldwide, we also compared the different approaches adopted in the studies, in order to detect possible fluctuations in the range of the obtained estimates.

## 2. Materials and Methods

### 2.1. Preliminary Considerations

This review was performed according to the Preferred Reporting Items for Systematic reviews and Meta-Analyses (PRISMA) [14]. Approval by an Ethical Board was not required because all collected and analyzed data were retrieved from previously published studies, and our study included no individual patient data. This systematic review was registered in the International Prospective Register of Systematic Reviews (PROSPERO, registration number: CRD42021278753).

### 2.2. Eligibility Criteria and Study Selection

A systematic literature search was conducted among articles published from inception to 1 March 2021 in order to identify studies evaluating the economic impact of cancer (in terms of direct and/or indirect costs sustained by individuals and/or by institutions due to the disease), in relation to attributable DALYs. All primary research contributions were considered eligible for inclusion if they provided economic analyses that included figures related to the above-mentioned metrics (whether computed, estimated or retrieved from available databases), related to cancer. In particular, any observational or experimental study, as well as any primary economic evaluation, that met these inclusion criteria was to be considered eligible. Systematic reviews and other review-type studies, as well as non-peer-reviewed papers and abstract-only documents, were excluded from the analysis. Furthermore, studies investigating DALYs averted due to tertiary prevention were ruled out, considering that our analysis was focused on the costs and DALYs related to actual reported disease.

### 2.3. Literature Search

A comprehensive search strategy (Appendix A) developed by a skilled researcher with a background in economics involved searching the following electronic bibliographic databases: PubMed, Scopus, Ovid Embase and Web of Science. Furthermore, potential additional studies were sought by “hand-searching” references from the selected articles (i.e., snowball searching [15]).

### 2.4. Outcomes

The primary outcome assessed was the cost-per-DALY ratio. However, as this outcome is not always provided by this kind of study, costs and DALYs related to the disease were considered as possible surrogate outcomes that were useful to compute the cost-per-DALY ratio for the quantitative synthesis. Additional outcomes included components of disease-related expenditure (i.e., direct and indirect costs).

### 2.5. Data Extraction

Each title and abstract was independently screened by two of the authors. Full texts of all the articles identified as relevant were then assessed for eligibility. Two authors, independently and in duplicate, identified studies that met inclusion criteria and extracted data through a predetermined data collection form (Appendix A). A consensus on extracted data was reached by discussion, and disagreements were sorted out through arbitration by a third author.

Information extracted from the studies included the publication year, country, years in which cost analyses and DALY computations were performed, aim, scope (national, international or global), study design, number of included patients/simulations, currency (and corresponding year), presence/absence of age-weighting and/or discounting (and corresponding parameters, where present), data sources for costs and DALYs, and perspective (societal, governmental or healthcare system-based).

Given that no ready-to-use data about the cost-per-DALY ratio was available in any of the eligible studies, crude estimates about costs (both direct and indirect) and DALYs of the relevant cancer were also extracted, along with their 95% confidence intervals (CIs) where available. For studies with data specifically referring to a country (or a group of countries), the GDP per capita of those countries (in the corresponding year) was also retrieved from the World Bank Data Catalog [16]. Hence, the cost-per-DALY ratio was computed as a crude ratio between total costs and DALYs ascribable to the disease.

In view of the different methodologies used for the economic evaluation, articles were divided into two groups: the first group encompassed studies where cost and DALY estimates had been extracted independently, and not based on previous estimates of the cost-per-DALY ratio itself; the second group encompassed studies that opted for a reverse approach and calculated costs based on DALYs. Only the first group was used for the quantitative synthesis of the cost-per-DALY ratio, so as to estimate this parameter by using directly retrieved data. In contrast, articles of the second group were used as a comparison term for the projection of possibly determined differences in estimates.

### 2.6. Quality Assessment

The quality of included studies was evaluated using the Quality of Health Economic Studies (QHES) instrument, a tool specifically developed [17] and validated [18] for economic evaluations [19], which yields a score ranging from 0 (lowest quality) to 100 (highest quality). Since two of the questions were not suitable for the risk-of-bias assessment of the included studies, the maximum obtainable score was correspondingly 86 points, thus the continuous scale was categorized into four groups: extremely poor-quality (0–21), poor-quality (22–43), fair-quality (44–65) and high-quality (66–86).

### 2.7. Statistical Analysis

For each study, average estimates were retrieved for both costs and DALYs, along with their 95% confidence intervals (CIs) where available. In the absence of reported uncertainty measures, the 95% CIs were estimated by applying a Monte-Carlo simulation method, with a number of simulations equal to the population (or number of virtual simulations) from which data were extracted [20].

All cost-related data were converted to US Dollars (USD) according to the Purchasing Power Parities (PPP) provided by the Organization for European Co-operation and Development (OECD) [21] and subsequently discounted to 2018 USD (as the most recent evidence was based on 2018 data) through an on-line inflation calculation tool [22] based on the US Consumer Price Index Data [23].

As the meta-analytic estimate was based on a ratio (between costs and DALYs), logarithmic transformation of the cost-per-DALY ratio was performed prior to CI estimation and meta-analytic synthesis: this process was intended to deal with the a priori expected skewed distribution of this parameter, as commonly acknowledged for ratio distributions such as odds ratios and relative risks [24,25].

The R package “meta” (version 5.0-0) was used for meta-analytic computation and plotting [26]. As required when high heterogeneity is expected a priori [27] (here data were drawn from different countries, and thus from different contexts), the analysis was performed using a random effects model according to DerSimonian and Laird [28]. The synthetic result was reported as an estimate of the cost-per-DALY ratio (equal to the exponential of the meta-analytic mean of the logarithms, computed through the “metamean” function), along with its 95% CI (evaluated through the standard normal distribution method). Study weights were generated using the inverse variance method. Heterogeneity among studies was assessed using the I^2^ statistic (threshold level for significant heterogeneity: ≥50%) and the chi-squared test for homogeneity (significance level for heterogeneity: *p* < 0.1) [29].

Since one of the studies was conducted within a different economic context, a post-hoc sensitivity analysis was conducted through computation of the average cost-per-DALY ratio by excluding that study. Moreover, the quantitative synthesis was also performed by considering the ratio between the obtained cost-per-DALY and the corresponding GDP per capita, in order to allow for more precise comparisons between studies since the economic value of DALYs would be obviously affected by the size of the income and GDP per capita.

Subsequently, the same analysis was conducted by considering indirect costs only, which are more closely related to income and productivity losses, and therefore to the GDP itself. For these further analyses, meta-analytic synthesis was conducted and reported as described above for the primary investigation. All statistical analyses were performed using the statistical software R (R Foundation for Statistical Computing, Vienna, Austria, version 4.0.5) [30].

## 3. Results

### 3.1. Study Selection

The study selection process is detailed in Figure 1. Of the 1156 records obtained after deduplication, 1079 were screened out as they were either review papers or studies that did not include both cost and DALY analyses, or alternatively simple posters or conference abstracts. Among the 77 articles selected by the screening process, 21 were non-relevant documents and 47 were related to studies about costs per DALY averted by cancer prevention (e.g., vaccination or screening) or therapy, and not the costs per DALY of the disease itself.

Furthermore, one study [13] was potentially eligible according to inclusion criteria but was excluded as it performed a simple scoping analysis based on DALYs and GDP per capita without providing clear specification of the data sources and methodologies deployed. Eventually, one study [11] was not included because it failed to provide figures feasibly allowing for the computation of the cost-per-DALY ratio (i.e., both absolute or standardized per 100,000 persons).

As a result, seven studies were included in this review, of which five [31,32,33,34,35] were also eligible for quantitative synthesis since they provided directly retrieved cost and DALY estimates. In contrast, two studies [36,37] provided only rough estimates of the costs starting from DALYs and pre-determined parameters (per-capita GDP or value of a statistical life, VSL) and therefore were kept as a comparison term for estimate difference projections.

### 3.2. Characteristics of Included Studies

The seven studies included in this review were published between 2012 and 2021. Six of these studies [32,33,34,35,36,37] adopted a societal perspective, and only one [31] chose a healthcare system perspective. Characteristics of all studies are summarized in Table 1 and Table 2.

The five studies retrieving costs and DALYs from independent sources were based on national data from four countries (Korea, Mexico, the Netherlands and Portugal). These studies basically aimed at assessing health and economic burden of cancer in the relevant country: three of them [31,32,33] were longitudinal retrospective studies based on data from different time spans (i.e., 5–20 years), one [34] was designed as a prevalence study and one [35] was based on a virtual prospective cohort. Data sources included: for direct costs, national pension institutes, health authorities and insurance corporations; for indirect costs, central statistics bureaus and national ministries; and for DALYs, databases from (and/or software based on) surveys developed by supranational institutions (i.e., World Health Organization and European Union), national institutes for statistics or cancer registries.

The other 2 studies were economic evaluations (in one case [37] nested into a prevalence study) on a wider scale, either international or worldwide. They basically aimed at estimating the economic burden of one or more cancers, and they were based on DALY-related data coming from the Global Burden of Disease studies and from other authorities, such as the Institute for Health Metrics and Evaluation and the World Health Organization. In these studies, costs were estimated multiplying DALYs by pre-determined parameters, i.e., GDP per capita [36] and Value of Statistical Life (VSL) [37], respectively.

### 3.3. Quantitative Analyses

Considering the five studies independently performing cost and DALY analyses on a national scope, burdens of cancer included in each analysis differed appreciably among the studies under investigation, with health burdens ranging from 8.93 to 679 thousand DALYs, and relevant costs from USD 94.23 to USD 3515 million. This corresponded to a cost-per-DALY ratio spanning from around USD 5200 to USD 25,800 for each DALY ascribable to the disease (Table 3a).

The meta-analytic estimate of this cost-per-DALY ratio showed that this parameter could be reasonably quantified in USD 9150 (95%CI: (5560–15,050)) 2018 USD per DALY ascribable to the disease (Figure 2a). As four of the studies [31,32,33,34] included in this quantitative synthesis were conducted in high-income countries, while one [35] was set in a country with a different economic framework (upper-middle income), the same analysis was conducted excluding the latter study, resulting in a substantially unchanged estimate (USD 10,070 USD/DALY, 95%CI: (5730–17,660), Appendix A).

In all these five studies, this estimate was much lower than the numerical value of the GDP per capita (Table 3a). Meta-analysis conducted on the ratio between the obtained cost per disease-related DALY and the corresponding GDP per capita showed that the cost per DALY is on average 31.6% (95% CI: (23.6–42.1)) of the national GDP per capita. This analysis also pointed out the heterogeneity (I^2^ = 91%) among different geographical and socioeconomic contexts, as further shown by the post-hoc subgroup analysis (*p* < 0.0001) conducted by separating the two studies sharing the same origin (i.e., Korea) [32,34], where this ratio was as low as 21.8% (95% CI: [19.6–24.2]), from the others located in different settings, where the same parameter was estimated around 45.2% (95% CI: [34.7–55.7], Figure 2b).

A similar meta-analytic computation was performed on the ratio between indirect costs and DALYs, which showed that every DALY ascribable to cancer is responsible for an average indirect expenditure around USD 3890 (95% CI: (2570–5880)) 2018 USD (Figure 3a). Considering high-income countries only, these average indirect costs rise up to USD 5150 (95% CI: (3450–7680)) 2018 USD (Appendix A).

Meta-analysis conducted on the ratio between the obtained indirect cost per disease-related DALY and the corresponding GDP per capita showed that the indirect expenditure is on average 14.1% (95% CI: (11.9–16.8)) of the numerical value of the national GDP per capita (Figure 3b). Again, remarkable heterogeneity (I^2^ = 96%) could be identified between different geographical and socioeconomic contexts.

### 3.4. Quality Assessment

The quality and risk-of-bias assessment of studies included in this review is reported in the Appendix A. One study [37] turned out to be of poor quality, four studies [31,32,34,36] were classified as fair and two studies displayed valuable methodological qualities [33,35]. Interestingly, two of the included studies [32,37] lacked precise reporting of the currency and pricing date. Moreover, one study did not adopt a society-based perspective, focusing on direct costs only [31].

## 4. Discussion

This study was intended to provide an insight into the currently available literature about the cost and DALY analyses related to cancer. Moreover, in view of the various approaches adopted by the studies, this review also aimed at investigating the extent to which diverse estimation and computation methodologies might lead to differences in magnitude among the obtained estimates, thereby resulting in possibly disparate advice provided to policy-makers.

Despite the recent efforts to quantify the economic loss due to cancer, few papers examining both costs and DALYs have been published. Several studies evaluated either DALYs [44,45] or costs [46,47,48], but there is still a paucity of simultaneous analyses of these two parameters in relation to cancer.

The present systematic review and meta-analysis of economic evaluations showed a pooled cost-per-DALY ratio of around 9000 2018 USD per DALY accountable to cancer. Results also displayed that the total cost per DALY equals approximately one third of the national GDP per capita. Considering only indirect costs, the pooled ratio was around 4000 2018 USD. Relating the indirect costs-per-DALY ratio to the national GDP results in indirect expenditure per DALY equaling one seventh of the annual country’s economic output per person.

The quality of the studies reviewed was heterogeneous. There were differences in DALY computation with regard to the use of discounting since three [33,35,37] of the seven studies adopted a discount rate, even if not recommended by current standards [49]. In fact, supranational bodies such as the WHO and evidence in the scientific literature advised against the use of discounting and weighting in DALY computation as no intrinsic reason appeared to provide grounds for decreasing the value of a year of health simply because in the future [50], especially in the context where DALYs are explicitly used to quantify loss of health rather than the social value of this loss [51,52].

However, the meta-analysis clearly demonstrates the potentially substantial differences among estimates provided by studies conducted through different methodologies. Costs evaluated a posteriori after data collection and model building showed that the overall cancer-related indirect expenditure and productivity losses per DALY ascribable to the disease were much lower than the numerical value of the GDP per-capita (Figure 3), and the same held for the total expenditure (i.e., also including direct costs, Figure 2). This implies that the actual estimates of global costs of disease may be quite different from the “economic value” obtained multiplying DALYs by the GDP per capita. For example, the fact that cost estimates would be cut by around 70% (Figure 2b) implies that, if a posteriori computed parameters were used, the global cost estimate forecast by John et al. [36] would be reduced from one billion to 300 million 2018 USD.

This might be explained by the fact that GDP measures the monetary value of all final goods and services produced in a country in a given period (generally a year). Consequently, GDP is composed of goods and services produced for sale in the market and includes some non-market production (e.g., defense or education services provided by the government) [53], which may entail an excess computation of costs in relation to the actual economic burden of the disease when the GDP is taken as a predetermined proxy of the cost per DALY.

The remarkable differences highlighted by this analysis cast doubts on the reliability of estimates obtained by valuing the monetary burden from simple analyses based on DALYs and GDP per capita, a methodology widely used not only for cancer [13] but also in many other sectors such as infectious disease [54], injuries [55] and maltreatment-related illnesses [56].

A limitation of our calculation lies in the fact that meta-analytic estimates were obtained from studies conducted in high-income countries only: therefore, the results in terms of cost-per-DALY ratio may not be generalizable to low-income or low-middle-income countries (LMICs). However, the magnitude of the difference in global estimates according to the methodology is unlikely to be affected by these latter countries, since their relative contribution to global costs is limited (Table 3b).

Another study used the VSL as a proxy of the cost per DALY: the VSL can be defined as the monetary value of a mortality risk reduction that would prevent one statistical death [57,58]. When the tradeoff values are based on choices in market contexts, the VSL can be adopted as an indicator of the marginal cost of enhancing safety and the population’s willingness to pay for mortality risk reduction [59].

The use of the VSL as a proxy of the cost per DALY has already been applied to genetic [60], occupational [61] and infectious diseases [62]. However, it is interesting to notice that figures obtained by Ranganathan et al. [37] using the VSL methodology appeared to provide a cost-per-DALY estimate absolutely comparable to the value of the GDP per capita, with no appreciable difference in the magnitude of estimated costs: in fact, the ratio between VSL and GDP per capita ranged between a minimum of 0.823 (in Nepal, 786/955) and a maximum of 1.058 (in Pakistan, 1521/1437, see Table 3b for detailed figures). Thus, it is likely that, as previously observed for the case of GDP per capita, even using the VSL may equally lead to overestimating total costs of DALYs.

In light of the above, some implications emerge for both researchers and policy-makers. First of all, the availability of reliable estimates on the cost of cancer is absolutely required in order to steer targeted choices at the several levels of the decision-making process and to accurately allocate resources. For this reason, further studies are needed to provide a reasonable cost-per-DALY parameter that can be used to estimate the costs of cancers starting from relevant DALYs, similarly to the approach already adopted for different sectors [63], where previously published cost-per-DALY estimates [64,65] were considered and applied instead of fixed parameters like GDP or VSL. This would imply more challenging research but would also lead to a clearer understanding of the actual economic burden ascribable to cancer.

Subsequently, more precise computations of this burden would allow policy-makers to quantify the number of resources deployed in relation to DALYs and therefore to conversely evaluate the economic benefits resulting from the reduction of these pathologies [66,67]. As a result, this would also help policy-makers to assess the need for improving healthcare policies in terms of primary prevention, screening programs and early diagnosis of cancer diseases [68].

The present findings should be assessed in light of this study’s weaknesses and strengths. Along with the systematic search and analysis methodology, widely referenced in the scientific literature, this study is, to our knowledge, the first attempt to provide a quantitative synthesis of data inherent to costs per DALY ascribable to cancer. However, some limitations are present: firstly, the systematic review took into account studies investigating several types and/or a pool of cancers: the limited sample size prevented us from performing any subgroup or stratified meta-analysis for different cancer types. The presence of studies investigating different cancers—from breast cancer to multiple myeloma, thus with different treatment pathways and prognoses, which would result in different costs per DALY—is probably one of the reasons for the substantial heterogeneity of the obtained results. Moreover, another source of heterogeneity was represented by the fact that the retrieved original studies were conducted in different countries and therefore in different social and economic contexts: trying to overcome this limitation was the main reason for which the cost-per-DALY ratio was also analyzed in relation to GDP per capita, so as to lower as much as possible the biasing impact of income differences on the pooled estimate. However, an important fact to be noted is that, although the heterogeneity of results should be carefully considered, the differences between GDP-based evaluations and a posteriori calculated data are always wide since even the highest estimates provided by the studies for the cost per DALY did not exceed 53.4% of the respective GDP per capita.

Moreover, studies included in the quantitative synthesis were originally performed in different times, from 2008 to 2018; thus, they refer to—and their results might consequently depend on—the time period of data origination. Although currency discounting has been performed according to the US dollar inflation, other factors (e.g., inflation in costs related to health care and/or productivity losses) may affect the accuracy of the obtained estimates. Detailed information about this issue could not be easily accessed; however, given that similar estimates were obtained from comparable contexts in different times (see, for instance, Oh et al. [34] and Noh et al. [32], Figure 2b), we might speculate that this further parameter is unlikely to bias the estimate to a remarkable extent, if proper currency discounting and conversion procedures are applied.

Further studies are needed to investigate the extent to which these estimates, computed for a specific country, can be generalized to other settings, both in absolute terms and in relation to wealth and income indicators such as the GDP per capita. This holds particularly for low- and low/middle-income countries, where detailed research still has to be conducted on the costs and DALYs of cancer. Eventually, an additional line of research could involve assessing how the adoption of different cost-per-DALY computations may affect the prioritization of interventions.

## 5. Conclusions

Cancer represents a major health issue, concerning both the clinical burden (in terms of morbidity and mortality) and the consequent economic implications. With regard to the latter, DALYs are often used to measure the burden of disease since they are a compound unit encompassing both disability and mortality, but substantial heterogeneity occurs when they are translated to monetary value.

Each DALY due to cancer has shown to cost, on average, around 9000 2018 USD in high- and upper-middle income countries, although this computation can be strongly influenced by fluctuations depending on cancer type and other parameters (e.g., country, prices). Moreover, the cost per cancer-related DALY has been found to be, on average, 32% (95% CI: 24–42%) of the corresponding countries’ GDP per capita, which implies that the use of a priori established parameters, such as GDP or VSL, might lead to presenting rough estimates highly different (even threefold) from what emerges a posteriori, after directly retrieving figures and/or building models out of available data. Therefore, further research is needed to understand the actual value of DALYs connected to cancer, and to elaborate on reliable cost estimates that may provide reasonable ground to policy-makers for appropriate decision-making.

## Figures and Tables

**Figure 1 ijerph-19-04862-f001:**
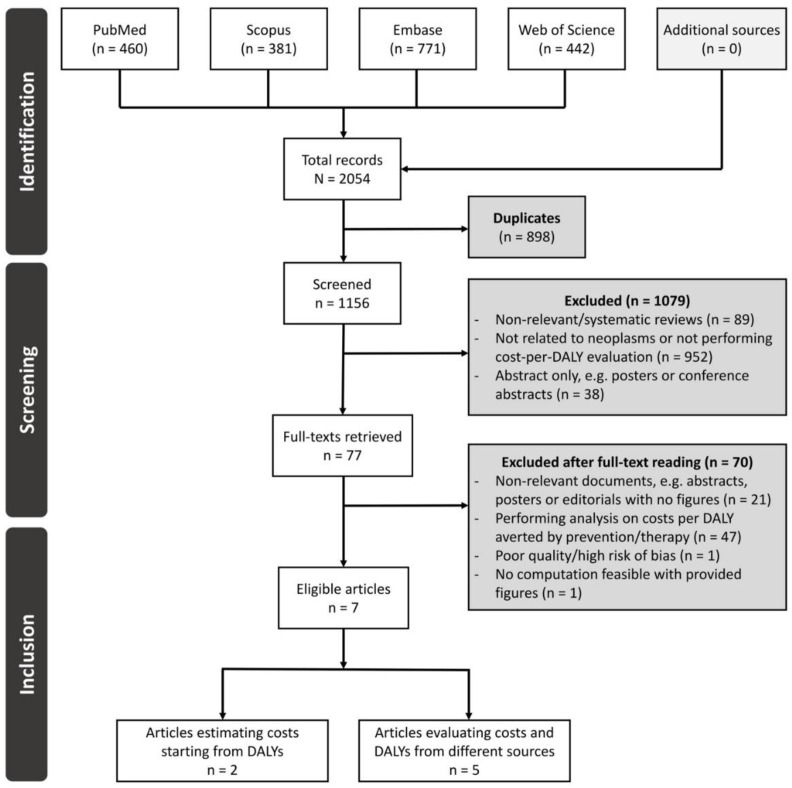
Study selection flow-chart according to the PRISMA Standard.

**Figure 2 ijerph-19-04862-f002:**
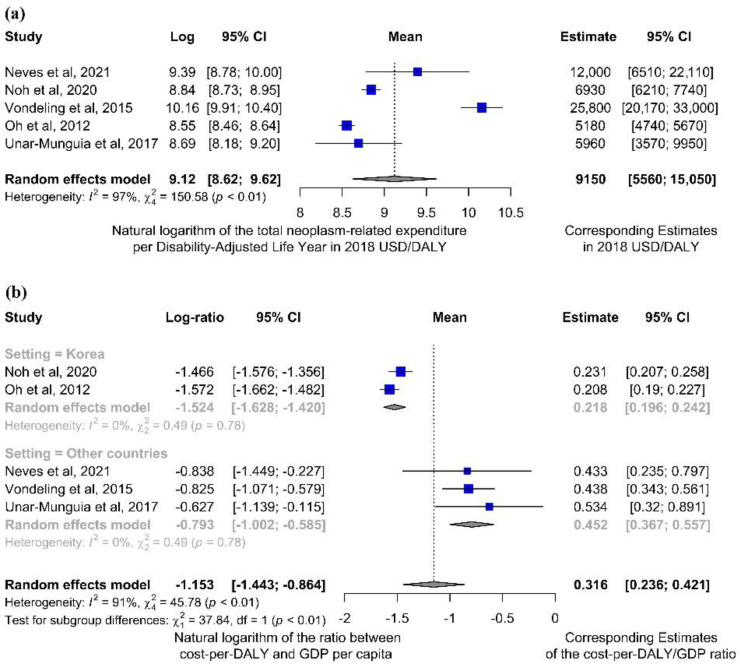
Forest plots for outcomes related to the cost-per-DALY ratio, according to meta-analytic computations described in the Methods. A meta-analytic estimate of the average cost per DALY ascribable to cancer (and its 95% CI), performed considering all studies eligible for the quantitative synthesis [31,32,33,34,35] (**a**). Expenditures per DALY were also considered in relation to the corresponding GDP per capita; (**b**) given that two studies were set in the same context (Korea) [32,34], a *post-hoc* subgroup analysis was also performed by separating them from other settings.

**Figure 3 ijerph-19-04862-f003:**
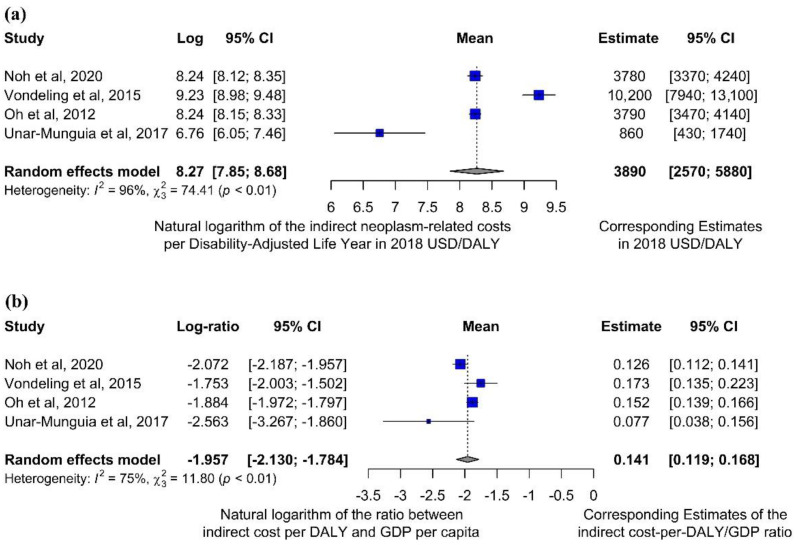
Forest plots for outcomes related to the ratio between indirect costs and DALYs, according to meta-analytic computations described in the Methods. A meta-analytic estimate of the average indirect cost per DALY ascribable to cancer (and its 95% CI), performed considering all studies reporting indirect cost data [32,33,34,35] (**a**). Indirect expenditures per DALY were also considered in relation to the corresponding GDP per capita (**b**). One study [31] was not included in this computation due to absence of indirect cost data.

**Table 1 ijerph-19-04862-t001:** The descriptive characteristics of included studies: the general characteristics of included studies. White rows indicate studies performing separate economic analyses related to costs and DALYs (included in the quantitative synthesis), whereas shaded rows contain studies that deduce costs starting from DALYs (not included in the meta-analysi).

1st Author, Year	Scope and Setting	Study Period	Study Aim	Study Type	Nr. Patients/Simulations	Perspective	Neoplasm Type
Neves et al., 2021	National: Portugal	2014–2018 *	To assess the burden and costs of multiple myeloma in Portugal	Longitudinal retrospective study	1941	Healthcare system	Multiple myeloma
Noh et al., 2020	National: Korea	2006–2015 *	To compute lung cancer burdens related to radon	Longitudinal retrospective study	69,168	Societal	Lung cancer
Vondeling et al., 2018	National: Netherlands	1995–2014 *	To report the total health and economic burden ascribable to breast cancer	Longitudinal retrospective study	73,261	Societal	Breast cancer
Oh et al., 2012	National: Korea	2008	To quantify the health and economic burden of smoking-related cancers in Korea	Cross-sectional study	NA ^†^	Societal	11 major smoking- related cancers
Unar-Munguia et al., 2017	National: Mexico	2012	To estimate breast cancer’s lifetime economic and disease burden, attributable to suboptimal breastfeeding practices	Simulated prospective study	100,000 (virtual)	Societal	Breast cancer
John et al., 2010	Worldwide: 205 countries	2004 (country- level data), 2008 (data by GNI group)	To estimate global economic losses due to cancers, starting from the economic burden expressed in DALYs	Economic evaluation	NA ^§^	Societal	17 types of cancer
Ranganathan et al., 2020	International: Low-middle-income countries	2005–2015 *	To estimate breast cancer survival trends and to quantify the economic burden of breast cancer in low-middle-income countries	Cross-sectional study and economic evaluation	NA ^§^	Societal	Breast cancer

* For these studies, reliable data on costs and DALYs were available for the last year only. Therefore, only these data were considered for the present study. ^†^ Not specified in the study. ^§^ Not applicable as the study was based on overall DALY data already provided in previously published studies.

**Table 2 ijerph-19-04862-t002:** Descriptive characteristics of included studies: economic parameters and data sources. White rows indicate studies performing separately economic analyses related to costs and DALYs (included in the quantitative synthesis), whereas shaded rows contain studies that deduce costs starting from DALYs (not included in the meta-analysis).

1st Author, Year	Currency (Year)	Age- Weighting	DALY Discounting	Data Source(s) for DALY Burden	Data Source(s) for Costs
Neves et al., 2021	EUR, 2018	NA	NA	**YLLs:** mortality data from European Cancer Information System, life expectancy data from the Portuguese Institute of Statistics **YLDs:** prevalence from Portuguese National Healthcare System hospitals, disability weight from WHO expert panels and GBD Study 2016	**Direct:** National tariffs, cost by diagnosis-related-groups, Iqvia databases **Indirect:** NA
Noh et al., 2020	USD, 2013 (for indirect costs) and 2017 (for direct costs)	NA	NA	**YLLs:** mortality data and life expectancy tables from Statistics Korea **YLDs:** population statistics and prevalence from Korean National Health Insurance database and cancer registry, disability weights from the literature [38]	**Direct:** National Health Insurance Service and Out-of-Pocket payments from the National Health Insurance Corporation survey **Indirect:** morbidity, unemployment or premature death from the Labour Statistics Bureau (Ministry of Health)
Vondeling et al., 2018	EUR, 2014	NA	1.5%	**YLLs:** Netherlands Life Expectancy Tables and mortality data from Netherlands Comprehensive Cancer Organization (IKNL) and National Cancer Registry (NCR) **YLDs:** incidence data from IKNL and NCR, disability weight from literature [39] and duration of disease from EUROCARE-4 project [40]	**Direct:** National Institute for Public Health and the Environment and Central Statistics Bureau **Indirect:** socio-economic data from the Central Statistics Bureau
Oh et al., 2012	USD, 2008	NA	NA	**YLLs:** Mortality data and specific frequency measures from Korean National Health Insurance Corporation database and the National Statistical Office **YLDs:** incidence data from Korean National Health Insurance Corporation database and the National Cancer Information Center, disability duration through DISMOD II software [41]	**Direct:** National Health Insurance Corporation records and 2009 national survey for out-of-pocket costs. Korea Health Panel survey for direct non-medical costs and caregivers’ costs **Indirect:** socio-economic data from the Ministry of Employment and Labour and cause of death from the National Statistical Office
Unar-Munguia et al., 2017	USD, 2015	NA	3%	**YLLs and YLDs:** computation provided by the DALY calculator (R software) [42] based on parameters from Global Cancer Observatory and Global Burden of Disease 2010	**Direct:** Mexican National Previdence Institute and Ministry of Health **Indirect:** socio-economic data from the National survey of Occupation and Employment
John et al., 2010	USD, 2008	NA	NA	**YLLs and YLDs:** Global Burden of Disease 2004 and 2008	**Estimated** as GDP * DALY
Ranganathan et al., 2020	USD, 2015	NA	3%	**YLLs:** incidence data from a variety of sources including WHO, censuses, vital registrations, and population-based cancer registries, life expectancy tables from Institute for Health Metrics and Evaluation **YLDs:** data from Institute for Health Metrics and Evaluation, Global Health Data Exchange	**Estimated** as VSL * DALY, where VSL was computed according to population and socio-economic statistics from literature [43]

DALY(s): Disability-Adjusted Life Year(s); DISMOD: DISease MODelling Software; EUR: Euros (€); GBD: Global Burden of Disease; GDP: Gross Domestic Product; USD: US Dollars (USD ); VSL: Value of a Statistical Life; WHO: World Health Organization; YLD(s): Year(s) Lived with Disability; and YLL(s): Year(s) of Life Lost.

**Table 3 ijerph-19-04862-t003:** Outcomes of included studies. (a) For studies separately performing economic analyses related to costs and DALYs, relevant figures are reported for both measures of the burden (first two columns): the cost-per-DALY ratio was computed (third column), and the table also reports the GDP per capita of the corresponding countries, as retrieved by the World Bank Data (last column). (b) For the remaining studies, assumed cost-per-DALY ratios were retrieved from the studies, along with DALY and derived cost estimates. All data were reported after discounting to 2018 USD.

(a) Studies performing separate economic analyses related to costs and DALYs.					
**1st Author, Year**	**Income Group**	**Costs (2018 USD mlns)**	**DALYs (Thousands)**	**Cost-per-DALY Ratio (a Posteriori) (2018 USD/DALY)**	**GDP per Capita (2018 USD)**
Neves et al., 2021	High	106.9	8.93	12,000	27,736
Noh et al., 2020	High	2460	355	6900	30,015
Vondeling et al., 2018	High	1666	64.6	25,800	58,846
Oh et al., 2012	High	3515	679	5200	24,949
Unar-Munguia et al., 2017	Upper-middle	94.23	15.8	6000	11,160
(b) Studies estimating costs based on disease burden (DALYs) and a priori assumed cost-per-DALY ratios.					
**1st Author, Year**	**Income Group**	**Costs (2018 USD mlns)**		**DALYs (thousands)**	**Cost-per-DALY Ratio (*a priori*)** **(2018 USD/DALY) ^†^**
John et al., 2010	All groups represented	High income: 853,000 Upper-middle income: 90,600 Low-middle income: 85,700 Low income: 14,700 **Total costs: 1044,000**		High income: 18,094 Upper-middle income: 8727 Low-middle income: 34,389 Low income: 21,645 **Total DALYs: 82,855**	High income: 47,100 * Upper-middle income: 10,400 * Low-middle income: 2500 * Low income: 680 *
Ranganathan et al., 2020	Low-middle	**Sub-Saharan Africa: 1964** **Southern Asia: 2090** India: 1419 Bangladesh: 153 Bhutan: 1.235 Nepal: 14.3 Pakistan: 473		**Sub-Saharan Africa: 604.9** **Southern Asia: 1294** India: 840.4 Bangladesh: 119 Bhutan: 0.439 Nepal: 18.2 Pakistan: 311	**Sub-Saharan Africa: 3250** * **Southern Asia: 1615 *** India: 1689 * (GDP per capita: 1701) Bangladesh: 1285 * (GDP per capita: 1323) Bhutan: 2813 * (GDP per capita: 2916) Nepal: 786 * (GDP per capita: 955) Pakistan: 1521 * (GDP per capita: 1437)

Legend. mlns: millions; USD: United States dollar; DALY: Disability-adjusted Life Years; and GDP: Gross Domestic Product. * The estimates were computed based on the reported data. ^†^ Where data is referred to a specific country, the corresponding GDPs per capita (retrieved from the World Bank Data Catalog [16]) were also reported.

## Data Availability

The data presented in this study are available on request from the corresponding author.

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
