# Peer review of "Cancers: What Are the Costs in Relation to Disability-Adjusted Life Years? A Systematic Review and Meta-Analysis"

_ijerph, 2022, doi:10.3390/ijerph19084862_

Round 1
Reviewer 1 Report
The manuscript should describe the criteria for inclusion of articles in the study. (describe Figure 1).
As the meta-analytic estimate was based on a ratio (between costs and DALYs), log-162 arithmic transformation of this parameter was performed prior to CI estimation and meta-163 analytic synthesis: this process was intended to deal with the a priori expected skewed 164 distribution, as commonly acknowledged for ratio distributions like odds ratios and rela-165 tive risks - this should be explain.
Conclusions are not aligned with results .
Author Response
We thank the reviewer for the valuable comments. Please, see the attached point-by-point response.

Reviewer 2 Report
This study aimed to assess the average cost per DALY of cancer by reviewing the currently available literature for the purpose of better estimating the economic burden of cancer in terms of costs and DALYs to inform policy makers.
The strengths of the study is the in depth analysis provided. The quantitative methods used are clearly indicated and, as such, could be easily replicated. The weaknesses are that there are few studies to compare and of those that are comparable, they are from high income countries and how they relate the DALY differed from each study to the extent that the authors had to interpret results substantially. Unfortunately, the principles they used in conducting this qualitative component of their research are not clear.
A major problem is that some references are missing and much of the research cited is not recent—this is also a problem for the studies investigated. Since the intention of this work is to act as a resource for developing policy, it is important that this paper be well supported by research and current (as should be studies evaluated); otherwise, this is a historical study, not one on which policy should be based.
There are a few problems with language and there is no clear distinction made between cancer and neoplastic disease. As a result, it is unknown whether the research the authors reviewed was only related to cancer or to all neoplastic diseases. This needs to be made clear in the body of the work.
It would have been helpful if the authors would have cited their references as expected and created their reference list as per MDPI style. This will have to be done before the paper is acceptable for publication.
Below is a line by line list of the suggested edits.
Line by line suggested edits
2 Given that the discussion of the paper relates to malignant neoplasms and neoplasm is not a keyword suggested for this paper by the authors, “neoplasm” should be changed to “cancer”.
2-3 Change “Neoplastic Disease: Which Costs in relation to Disability-Adjusted Life Years? A Systematic Review and Meta-Analysis” to “Cancers: A Systematic Review and Meta-Analysis of Costs in Relation to Disability-Adjusted Life Years”.
13 Change “neoplasms” to “cancers”.
15 Change “neoplasm” to “cancer”.
16 Change “approaches. We” to “approaches we”.
17 Change “neoplasm” to “cancer”.
25 Change “a priori calculated” to “calculated a priori”.
34-35 Change “(Sung et al., 2021)” to “[1]”.
36 Change “(Fitzmaurice et al., 2019)” to “[2}”.
36-39 Need a reference for this statement.
42 “(Brown, Lipscomb, & Snyder, 2001)” this reference is out of date. Please find a more recent reference and cite it as “[3]”.
42-45 Need a reference for this statement.
47 Change “(Mattiuzzi & Lippi, 2020)” to “[4]”.
49-50 Change “(C. J. Murray, 1994; Oostvogels et al., 2015)” to “[5,6]”. Only include the reference to Murray if this is a seminal work in the field. Otherwise, find a more recent reference.
52 Change “(Chao, 2020; C. J. L. Murray & Acharya, 1997)” to “[7,8]”. Only include the reference to Murray & Acharya if this is a seminal work in the field. Otherwise, find a more recent reference.
53-55 Delete “Premature deaths and disability from cancer result in substantial economic costs to the patients, their family and the community as a whole. Therefore, cancer-related DALYs cancer place an enormous burden on national economies.” this information has already been provided in the opening two paragraphs and is not relevant to the remainder of the paragraph.
57 “in 2008” this information is very out of date. Find a more recent reference.
58 Change “(American Cancer Society, 2010)” to “[9]” but find a more recent reference.
61 Change “(National Cancer Institute, 2019)” to “[10]”.
63-64 Change “(Schlueter, Chan, 63 Lasry, & Price, 2020)” to “[11]”.
66-68 “considering that currently 30-50% of cancers can be prevented by avoiding risk factors and by implementing the existing evidence-based prevention strategies” need a reference.
69-70 Change “(World Health 69 Organization (WHO), 2021)” to “[12]”.
76-77 Change “(Jimenez de la Jara et al., 2015)” to “[13]”.
79 Change “revising” to “reviewing”.
86 Change “(PRISMA)” to “Tables 1 and 2” and change “(Page et al., 2021)” to “[14]”. In the PRISMA checklist attached as a separate file, relabel “PRISMA 2020 Main Checklist” as “Table 1. PRISMA 2020 Main Checklist” and “PRIMSA Abstract Checklist” as “Table 2. PRIMSA Abstract Checklist”.
92 Why was the search conducted from inception rather than over the last five years? This is not a historical study. The authors are looking to provide information about costs. Costs beyond the last five years are not relevant.
93-94 Change “neoplasms” to “cancers” if neoplasms were searched rather than cancers, then all instances in this paper that mention cancers should be changed to neoplasms, including in the keywords. However, if this change is made, then the statistics mentioned in the introduction should be for neoplasms in general rather than just for cancers and new references must be cited.
98 Change “neoplasms” to “cancers” (unless all references to cancers are changed to neoplasms).
197 “snowball searching” need a reference.
105-106 Change “(strings reported in the Supplemental Material)” to “(See Table 3)” and relabel what has been referred to in the supplementary material as “Search strings” to “Table 3. Search strings”.
110 Change “this kind of studies” to “this kind of study”.
118 Change “(Supplementary Material)” to “(Table 4) and relabel what has been referred to in the supplementary material as “Data collection form” to “Table 4. Data collection form”.
127-128 Further explanation has to be provided regarding how interpretations were made that the information available pertained to cost-per-DALY ratio and how these crude estimates were made.
129 Change “neoplastic disease” to “cancer”.
130 Change “referred” to “referring”.
132 Change “(The World Bank, 2010)” to “[15]”.
139 Change “On the contrary” to “In contrast”.
143 Change “(Chiou et al., 2003)” to “[16]”.
144 Change “(Ofman et al., 2003) for economic evaluations (Walker et al., 2012)” to “[17] for economic evaluations [18]”. These references are out of date, please provide updated references.
155 Change “(Weir et al., 2018)” to “[19]”.
158 Change “(Organisation for European Co-operation and Development, 2020)” to “[20]”.
159 “and subsequently discounted to 2018 USD” if data are being used that are much older than five years ago then a rationale has to be provided for why the type of costs incurred are comparable to 2018. Have all other relevant parameters remained the same over that time? If not, what ones have changed and why does this change not impact the costs that are being calculated?
160 Change “(Coin News, 2021)” to “[21]”.
161 Change “(U.S. Bureau of Labor Statistics, 2021)” to “[22]”.
164 Change “a priori expected” to “expected a priori”.
166 Change “(Higgins, White, & Anzures-Cabrera, 2008; Katz, Baptista, Azen, & Pike, 1978)” to “[23, 24]”. Both of these are older references. See if there are newer ones. Unless Katz, Baptista, Azen, & Pike is a seminal work in the field, leave it out, it is much to old a reference.
168 Change “(Balduzzi, Rücker, & Schwarzer, 2019)” to “[25]”.
169 Change “a priori expected (Borenstein, Hedges, Higgins, & Rothstein, 2010)” to “expected a priori [26]”.
171-172 Change “(DerSimonian & Laird, 2015)” to “[27]”.
178 Change “(Higgins, Thompson, Deeks, & Altman, 2003)” to “[28]” and find a more recent reference.
179-180 Change “(Unar-Munguía, Meza, Colchero, Torres-Mejía, & de Cosío, 2017)” to “[29]”.
190 Change “(R Core Team, 2021)” to “[30]”.
193 Change “Fig. 1.” to “Figure 1”.
200 Change “(Jimenez de la Jara et al., 2015)” to “[13]”.
203 Change “(Schlueter et al., 2020)” to “[31]”.
206-208 Change “(Neves et al., 2021; Noh et al., 2020; Oh et al., 2012; Unar-Munguía et al., 2017; Vondeling, Rozenbaum, Dvortsin, Postma, & Zeevat, 2016)” to “[32-36]”.
209 Change “On the contrary” to “In contrast”.
209-210 Change “(John & Ross, 2010; Ranganathan et al., 2021)” to “[37, 38]”.
216-217 Change “(John & Ross, 2010; Noh et al., 2020; Oh et al., 2012; Ranganathan et al., 2021; Unar-Munguía et al., 2017; Vondeling et al., 2016) to “[37,33,34,38,35,36]”.
218 Change “(Neves et al., 2021)” to “[32]”.
219 Change “Tables S1-S2” to “Table 5 and Table 6” and relabel Tables S1 and S2 in the supplementary material to Table 5 and Table 6 and bring them into the main document.
222 Change “neoplastic disease” to “cancer” unless the studies were specifically about neoplastic disease in general and not cancer specifically.
223 Change “(Neves et al., 2021; Noh et al., 2020; Vondeling et al., 2016)” to “[32,33,36]”.
225 Change “(Oh et al., 2012)” to “[34]”.
225-226 Change “(Unar-Munguía et al., 2017)” to “[35]”.
232 Change “(Ranganathan et al., 2021)” to “[38]”.
238 Change ““(John & Ross, 2010)” to “[37]”.
239 Change “(Ranganathan et al., 2021)” to “[38]”.
246 Change “(Table 1a)” to “(Table 7a)”.
247 Relabel Table 1 as Table 7.
255-257 Please change the font size of this information to correspond to that of footnotes for tables as per the instructions to authors.
260 Change “Fig. 2a” to “Figure 2a”.
260 to 261 Change “(Neves et al., 2021; Noh et al., 2020; Oh et al., 2012; Vondeling et al., 2016)” to “[32,33,34,36]”.
262 Change “(Unar-Munguía et al., 2017)” to “[35]”.
265 Change “Fig. S1” to “Figure 3” and change the label of Figure S1 in the supplementary material.
267 Change “(Table 1a)” to “(Table 7a)”.
273 Change “(Noh et al., 2020; Oh et al., 2012)” to “[33,34]”.
275 Change “Fig. 2b” to “Figure 2b”.
278 Change “neoplastic disease” to “cancer”.
280 Change “(Korea)32,34” to “(Korea [33,34])”.
283 The studies should appear in the Figure in the order in which they are referenced in the paper.
294 Insert a blank line before “A”.
297 Change “Fig. 3a” to “Figure 4a”.
299 Change “Fig. S2” to “Figure 5”.
307 Change “neoplastic disease” to “cancer”.
309 Change “study31” to “study [31]” and check that it was actually [31] that is meant—it is more likely [32] that is intended.
314 Change “(Table S3). One study (Ranganathan et al., 2021)” to “Table 8. One study [38]” and change the label of the table in the supplementary material as well as bring the table into the main document.
315-316 Change “(John & Ross, 2010; Neves et al., 2021; Noh et al., 2020; Oh et al., 2012)” to “[37,32,33,34]”.
317 Change “(Unar-Munguía et al., 2017; Vondeling et al., 2016)” to “[35,36].
320 Change “(Neves et al., 2021)” to “[32]”.
323 Change “neoplastic disease” to “cancer”.
330-332 Change “(Melaku et al., 2018; Safiri et al., 2019) or Costs (Huang, Chen, Liao, Ko, & Hsiao, 2020; Luengo-Fernandez, Leal, Gray, & Sullivan, 2013; Yabroff, Lund, Kepka, & Mariotto, 2011)” to “[39,40] or Costs [41,42,43]”.
333 Change “neoplasms” to “cancers”.
336 Change “neoplasms” to “cancers”.
342-343 Change “(Ranganathan et al., 342 2021; Unar-Munguía et al., 2017; Vondeling et al., 2016)” to “[38,35,36]”.
344-345 Change “(World Health Organization (WHO) - Department of Data and Analytics, 2020)” to “[44]”.
348 Change “(Tsuchiya, 2002)” to “[45]” and provide a more recent reference than this one.
350 Change “(Anand & Hanson, 1997; Barendregt, Bonneux, & Van der Maas, 1996; C. J. L. Murray & Acharya, 1997)” to “[46,47,48]” but these references are far too old. Please find some recent references.
354 Change “a posteriori evaluated” to “evaluated a posteriori”.
355 Change “neoplasm” to “cancer”.
356-357 Change “Fig. 3” to “Figure 4”.
357 Change “Fig. 2” to “Figure 2”.
360 Change “Fig. 2b” to “Figure 2b”.
362 Change “(John & Ross, 2010)” to “[37]”.
367 Change “(Callen, 2020)” to “[49]”.
372 Change “neoplastic disease” to “cancer”.
373 Change “(Jimenez de la Jara et al., 2015)” to “[13]”.
374 Change “(Townsend, Greenland, & Curtis, 2017), injuries (Dalal & Svanström, 2015) and maltreatment-related illnesses (Mo et al., 2020)” to “[50], injuries [51] and maltreatment-related illnesses [52]”.
381 Change “Table 1b” to “Table 7b”.
384 Change “(Ashenfelter & Greenstone, 2004; Kniesner & Viscusi, 2019)” to “[53,54]”. Reference 53 is too out of date. Please find a more recent reference.
387 Change “(Bellavance, Dionne, & Lebeau, 2009)” to “[55]”.
389 Change “(Gong et al., 2021)” to “[56]” and “(Lebeau, Duguay, & Boucher, 2014)” to “[57]”.
390 Change “(Herrera-Araujo, Mikecz, & Pica-Ciamarra, 2020)” to “[58]”.
391 Change “(Ranganathan et al., 2021)” to “[59]”.
396 Change “Table 1b” to “Table 7b”.
400 Change “neoplasms” to “cancers”.
405 Change “(Miller et al., 2020)” to “[60]”.
406 Change “(Hirth, Chernew, Miller, Fendrick, & Weissert, 2000; Viscusi & Masterman, 2017)” to “[61,62]” but reference 61 is too old. Please find a more recent reference.
413 Change “(Coons & Craig, 2008; World Cancer Leaders’ Summit, 2014)” to “[63,64]”.
These references are too old. Please find more recent references.
415-416 Change “(Emmons & 415 Colditz, 2017)” to “[65]”. 422 Change “neoplasms and/or a pool of neoplasms” to “cancers and/or a pool of cancers”.
424 Change “neoplasm” to “cancer”.
424-425 Change “neoplasms” to “cancers”.
445 Change “neoplasms” to “cancers”.
450 Change “neoplasms” to “cancers”.
455 Change “a posteriori computed” to “computed a posteriori”.
457 Change “neoplasms” to “cancers”.
461-476 If this information is still needed once the supplementary materials are brought into the body of the text then the information should be provided as part of the headings for the figures and tables. Change all instances of “neoplastic disease” to “cancer”.
487-494 Please complete these sections. The information currently listed is from the MDPI template.
References
All references must be redone to correspond with the MDPI style as listed in the instructions for authors.
Author Response

(The authors gave the same response as above.)

Reviewer 3 Report
This paper is rather well-written but has a number of textual issues which I have tried to help resolve within the manuscript PDF that I have uploaded. The approach and conclusions are acceptable; however, they are anchored to the time period of data origination in the cited references. Although the time value of different currencies has been adjusted to the US dollar, it is hard to determine if inflation in healthcare costs and in the different economies has been anticipated. While an answer to this question may not be available in the material reviewed, it certainly will be relevant to policy-makers. I favor adding some estimation of this consideration to the discussion.

Author Response

(The authors gave the same response as above.)

Round 2
Reviewer 2 Report
Thank you to the authors for responding to each of the points made by this reviewer to the first version of this paper. The authors have either made appropriate changes or provided a sufficient reason why certain changes were not made.
There remain only a few changes to make at this point. These are indicated in the line by line suggested edits below
Line by line suggested edits.
32 Change “Health Policy” to “health policy”.
115 Please list the databases as they are ordered in Table S1.
282 Please reformat the right column of the last listing so that each entry appears on one line without a large space before “(GDP”.
274 Change “94.23 to 3,515 USD million” to “$94.23 to $3,515 million USD”.
275 Change “5,200 to 25,800 USD” to “$5,200 to $25,800 USD
283 The first part of the sentence appears to be missing. It most likely should be “Legend”.
289 Change “9,150” to “$9,150”.
295 Change “10,070” to “$10,070”.
312-324 Please expand the left and right margins of this figure to the left and right margins of the text.
329 Change “5,150” to “$5,150”.
340-341 Please expand the left and right margins of this figure to the left and right margins of the text.
374 Change “although not” to “not”.
476 Change “Oh et al and Noh et al” to “[34} and [32]”.
References
5 Please indicate how this text is accessed. Is it accessed online or is it a hard copy? if a hard copy, where is it published and what are the number of pages?
10 The reference is incomplete.
15 The reference is incomplete. Please indicate how it was accessed.
19 The reference is incomplete.
21 Please indicate the date accessed.
30 Please indicate how the reference was accessed.
39-42 These references are all incomplete.
49 Please list the number of pages.
51 Please list the number of pages.
Author Response
We thank the reviewer for the comments. Please, see the attached file for a point-by-point response.
